# Improvement in Metabolic Co-Morbidities after Implantation of CardioMEMS in Patients with Heart Failure with Preserved Ejection Fraction Phenotype

**DOI:** 10.3390/jcm10194308

**Published:** 2021-09-22

**Authors:** Amit Alam, Johanna Van Zyl, Navdeep Nayyar, Shelley Hall, Rita Jermyn

**Affiliations:** 1Baylor University Medical Center, Department of Advanced Heart Failure and Transplantation, Dallas, TX 75246, USA; Shelley.Hall@bswhealth.org; 2College of Medicine, Texas A&M University, Bryan, TX 77801, USA; Johanna.vanzyl@bswhealth.org; 3Baylor University Medical Center, Department of Cardiovascular Research, Dallas, TX 75246, USA; 4Department of Cardiology, St. Francis Hospital, Roslyn, New York, NY 10001, USA; Navdeep.nayyar@chsli.org (N.N.); Rita.Jermyn@chsli.org (R.J.)

**Keywords:** pulmonary artery pressure monitoring, CardioMEMS, metabolic syndrome, heart failure with preserved ejection fraction

## Abstract

Background: Heart failure with preserved ejection fraction (HFpEF) patients often have other comorbidities, including obesity, dyslipidemia, hypertension, and diabetes, comprising the metabolic syndrome. The impacts of hemodynamic monitoring via CardioMEMS on these co-morbidities remain unknown. Methods: A retrospective analysis of 29 patients with HFpEF (EF 45% or greater) and CardioMEMS was performed at a single center. Weight, body mass index (BMI), systolic blood pressures (SBP), high-density lipoprotein (HDL), triglycerides (TGL), hemoglobin A1C (HbA1c), and pulmonary artery diastolic pressures (PADP) were assessed at baseline and six months post-implant. Paired *t*-tests and the Wilcoxon signed-rank test were used, as appropriate, to test differences between time points. Results: These patients were 69% female, with a mean age of 73 years, and 62% had non-ischaemic cardiomyopathies (NICM). At the time of CardioMEMS implantation, average PADP was 20.1 mmHg ± 5.7, weight was 102.6 kg ± 22.7, BMI was 38.0 kg/m^2^ ± 8.3, SBP was 135 mmHg ± 19, HDL was 42.4 mg/dL ± 11.3, and median TGL was 130 mg/dL (100, 180). At six months we witnessed a decrease by 20.9% in PADP to 15.9 mmHg ± 5.8, (*p* < 0.001). In addition, the following was noted: weight decreased by 2.5% to 100.0 kg ± 23.2, (*p* = 0.006), BMI reduced by 2.6% to 37.0 ± 8.2, (*p* = 0.002), SBP decreased by 6.7% to 126 mmHg ± 16 (*p* < 0.001), HDL increased by 10.8% to 47 mg/dL ± 11.9 (*p* < 0.001), and TGL decreased by 15.4% to 110 mg/dL (105, 135) (*p* = 0.001). 62% of patients were diabetic with no significant improvements in HbA1C values at the 6-month follow-up. Conclusion: The utilization of CardioMEMS to optimize PADP results in an improvement in the comorbidities associated with the metabolic syndrome. Further studies are warranted to validate these findings and delineate clinical significance.

## 1. Introduction

Heart failure (HF) continues to be a leading cause of hospitalization and death in the United States [1,2]. The management of patients with HF with preserved ejection fraction (HFpEF) remains challenging, as drug studies have failed to show a statistical improvement in mortality [3]. Many of these patients have metabolic syndrome, which further increases their morbidity and mortality [4,5,6]. These metabolic impairments may represent novel targets to slow disease progression and impact the outcomes of patients with HF [7].

A pulmonary artery pressure hemodynamic monitoring system, CardioMEMS, has already shown to improve quality of life and decrease hospitalizations in patients with HFpEF [8]. It is unknown whether hemodynamic monitoring via CardioMEMS will improve the metabolic traits and comorbidities associated with HFpEF. Numerous cardiometabolic traits contribute to the metabolic syndrome. The definition of this is different across the literature [9,10,11]. In this study, we investigated the metabolic comorbid conditions comprising metabolic syndrome, including, obesity, dyslipidemia, hypertension, and diabetes, before and after CardioMEMS implantation, and we assessed these components’ association with clinical outcomes. Our primary endpoint was evaluation of improvement in the metabolic traits from baseline to 6 months post-implantation of CardioMEMS. Secondary endpoints included the association of pulmonary artery diastolic pressures with device-related adverse events, death, and re-admission rates.

## 2. Methods

This retrospective cohort study was conducted at a single institution between 2018 and 2020 and consisted of 29 consecutive patients with HFpEF. Patients with an ejection fraction ≥ 45% were included in the analysis. This study was approved by the St. Francis Hospital’s institutional review board, and a requirement for informed consent was waived due to the retrospective nature of the study. Metabolic parameters, including weight, body mass index (BMI), systolic blood pressure (SBP), high-density lipoprotein (HDL), triglycerides (TGL), and hemoglobin (HbA1c), in addition to pulmonary artery diastolic pressures (PADP), were obtained at time of implant and six months after CardioMEMS implantation. Pulmonary artery waveforms were reviewed, and readings at the end expiration were collected.

### 2.1. Metabolic Parameters

Baseline characteristics, metabolic labs (at baseline and 6 months ± 1 month), re-admissions and survival data were collected during the first 6-month post-CardioMEMS implantation. The metabolic traits collected were BMI, weight, SBP, HDL, TGL, and HbA1c. Waist circumference, fasting glucose levels, and microalbuminuria values were not collected, as these data were not available for all patients. Time points of baseline and around 6 months after CardioMEMS implantation were chosen to allow sufficient time for PADP stabilization and titration of medications, namely diuretics.

### 2.2. Statistical Analysis

Continuous variables are presented as mean ± standard deviation, or median (interquartile range) if skewed, and categorical variables are presented as frequency (percent). Paired *t*-tests, or Wilcoxon’s signed-rank tests for non-normal data, were used to assess changes from baseline to 6 months post-CardioMEMS-implantation, as appropriate, for the distribution of the variables. Normality of the paired differences were assessed using a Shapiro−Wilk test and inspection of the normal quantile-quantile plots. A two-sided *p*-value of 0.05 was considered statistically significant. All analyses were performed by using R (R Foundation for Statistical Computing, Vienna, Austria, version 4.0.2).

## 3. Results

### 3.1. Baseline

Twenty-nine New York Heart Association (NYHA) class-three HF patients with an ejection fraction of 45% or greater were included in our study. Baseline characteristics included a mean age of 73 years, and 69% of patients were female (Table 1). The pre-implantation metabolic profile and hemodynamic data are also displayed (Table 2).

### 3.2. Trend in Metabolic Labs

Compared to baseline, at the six-month follow-up post-CardioMEMS-implantation, a 20.9% reduction in PADP occurred and a statistically significant improvement in weight, BMI, SBP, HDL, and TGL (all *p* < 0.01) (Table 2). A total of 62% of the patients were diabetics, and at the 6-month follow-up, the median change in HbA1c did not improve statistically.

### 3.3. Secondary Analysis

No adverse events related to the device implantation occurred. Of the 29 patients, none required dialysis and, in fact, CKD stage improved significantly (*p* = 0.008), with 41% of patients improving by at least one class. A significant improvement in NYHA classification from class III to II was observed in 31% of patients (*p* = 0.003), which was mimicked by a reduction in BNP by 55.7% (*p* = 0.002). There were no deaths in our study, and only 17% (*n* = 5) of patients were re-admitted over the 6-month follow-up period for HF exacerbations. None of the patients were lost to follow-up. (Table 3)

### 3.4. Medication Usage

In our study, we collected medication treatment data of hyperlipidemia, hypertriglyceridemia, and diabetes mellitus. All patients had diuretics adjusted based on pulmonary artery diastolic pressure goals. There were no changes noted in the medications used for treatment of hyperlipidemia and hypertriglyceridemia within the six months of follow-up. Moreover, there were no changes to medications for the treatment of diabetes mellitus. Diuretic adjustments, mostly in the form of loop diuretics were made one or two times per week, on average, for the first 6–8 weeks following implantation. After this initial period, loop diuretics were adjusted 0 or 1 times per week for the remainder of the study duration.

## 4. Discussion

HF is associated with metabolic incompetence and this study confirms that patients with HFpEF have some degree of metabolic derangement [4,5,6,10]. It is unknown whether the metabolic dysfunction is a consequence of HF or a contributing factor to HFpEF. In this study, we assessed changes in metabolic parameters after CardioMEMS implantation in consecutive patients with HFpEF and whether metabolic dysfunction post-implantation is improved. Our study finds that metabolic comorbidities improved in parallel as the PADP improved. In our study, a decrease in PADP was observed in 23 of 29 patients with the remaining six unchanged at 6 months post-implantation of CardioMEMS. This resulted in an improvement in PADP of 20.9% to 15.9 mmHg. In concert, SBP improved by 6.7% to 126 mmHg. Weight decreased by 2.5%, resulting in a reduction in BMI of 1 kg·m^2^ over a 6-month follow-up period. Finally, the mean HDL level improved by 10.8% to 47 mg/dL, and median triglycerides decreased by 15.4% to 110 mg/dL, despite medications used for treatment of hyperlipidemia and hypertriglyceridemia remaining the same during the study period. A low readmission rate for HF exacerbations without any deaths nor need for hemodialysis, and a significant improvement in NYHA class, BNP levels, and CKD stage, were observed.

Patients with HFPEF often have more comorbid metabolic issues than patients with HFREF. Their average BMI is higher, and they more often have DM and HTN [12]. Obesity is a known risk factor for development of HF, and it is postulated as a significant contributor to systemic inflammation, leading to myocardial remodeling with interstitial fibrosis and resultant HFpEF [13,14]. Furthermore, dyslipidemia results in pro-inflammatory, pro-oxidative, and pro-fibrotic effects, contributing to HF [15]. Due to a plethora of many systemic comorbidities, patients with HFpEF have elevated inflammatory cytokines, chemokines, and adhesion molecules. [16] Animal and human studies have already shown that restoration of normal cardiac hemodynamics results in improved end-organ function and decreased inflammation, and favorably affects metabolic pathways in the periphery and myocardium [11,17]. Accordingly, our study further supports and suggests that the optimization in PADP improves the risk factors for HFpEF and the comorbidities, as they relate to the metabolic syndrome.

With no changes in medications for hyperlipidemia, hypertriglyceridemia, or diabetes mellitus from baseline to 6-months, it is interesting to note that the only metabolic comorbidity that did not improve significantly was HbA1C, which remained similar at 6 months compared to baseline (6.5 vs. 6.8%). Patients with HFpEF on statin therapy are known to have less oxidative stress and cardiomyocyte hypertrophy. Studies are also suggestive that statins have a pleiotropic anti-inflammatory effect that will attenuate some of the inflammatory markers active in HFpEF [18]. The same is not appreciated for medications targeting diabetes management [18]. In our study, 76% percent of patients were on statin therapy, while only 41% of patients were being treated for diabetes. Additionally, with a HbA1C less than 7.0, many patients may not have been on any medicinal therapy at the discretion of their primary care provider, with attempts to improve lifestyle modifications. Sodium glucose co-transporter 2 inhibitors are known to have anti-inflammatory effects; however, none of the patients were on this medication in this 6-month follow-up, as they were not approved for HFpEF at the time of this study. Perhaps there is a synergistic effect of pairing medications that have an anti-inflammatory effect or in lifestyle modifications and optimization of pulmonary artery pressures that needs to further be investigated. Overall, the findings in our study are hypothesis-generating in the absence of a control group.

## 5. Limitations and Future Directions

Our study is also a retrospective analysis with a relatively small sample size. We enrolled 29 consecutive patients with HFpEF, and none were lost to follow-up for the duration of the study. We also were not able to collect all of the markers traditionally used for the metabolic profile, namely waist circumference, microalbuminuria, and fasting glucose. Nonetheless, our study results were sufficient to detect important improvements from baseline to six months post-CardioMEMS-implantation in hemodynamic and metabolic profiles. We also realize that cardiometabolic disease is known to disproportionately affect different ethnic demographics [2]. Our study was conducted in a Single Center in Roslyn, New York, and according to the 2010 Census, only 2.2% of the population in this town is African American. As a result, our study did not have enough power to perform race-specific analysis, with all of the participants in our study being Caucasian. Larger sample sizes of patients in multicenter randomized prospective studies are needed to assess whether improvement of PADP through CardioMEMS monitoring results in improvement of metabolic comorbidities associated with the HFpEF patient population.

## 6. Conclusions

Metabolic dysfunction is highly prevalent in patients with HFpEF. Improvement in the hemodynamics profile is associated with improvement in the metabolic comorbidities seen in this patient population. Further studies are warranted to validate these preliminary findings and delineate clinical significance on a patient’s morbidity and mortality.

## Figures and Tables

**Table 1 jcm-10-04308-t001:** Baseline characteristics.

Baseline Characteristics *	
Age, years	73 ± 12
Female	22 (69%)
Caucasian	29 (100%)
Creatinine (mg/dL)	1.3 (1.0, 1.8)
NYHA Class 3	29 (100%)
HF etiology, NICM	18 (62%)
History of Hypertension	19 (66%)
History of Pulmonary Hypertension	24 (83%)
History of Atrial Fibrillation	11 (38%)
History of Sleep Apnea	10 (34%)
History of Smoking	3 (10%)
History of COPD	3 (10%)
Diabetes Mellitus Medication	12 (41%)
Cholesterol Medication	22 (76%)
eGFR (mL/min/1.73 m^2^)	38 (21, 54)
CKD Stage	
2	5 (17%)
3A	4 (14%)
3B	10 (35%)
4	7 (24%)
5	3 (10%)
BNP (pg/mL)	280 (150, 480)

* Variables are summarized as the mean ± standard deviation or median (IQR) if continuous, and frequency (%) if ordinal or nominal; NYHA: New York Heart Association; HF: heart failure; NICM: non-ischemic cardiomyopathies; CKD: chronic kidney disease; BNP: B-type Natriuretic Peptide; COPD: chronic obstructive pulmonary disease.

**Table 2 jcm-10-04308-t002:** Changes in hemodynamics and metabolic traits. Paired *t*-tests or Wilcoxon’s signed-rank tests were used to assess changes from baseline.

Hemodynamic and Metabolic Profile *	Baseline Values	At Six-Month Follow-Up	% Change	*p*-Value
Pulmonary Artery Diastolic Pressure (mmHg)	20.1 ± 5.7	15.9 ± 5.8	20.9	<0.001
Systolic Blood Pressure (mmHg)	135 ± 19	126 ± 16	6.7	<0.001
Weight in kilograms (kg)	102.6 ± 22.7	100.0 ± 23.2	2.5	0.006
Body Mass Index (kg/m^2^)	38.0 ± 8.3	37.0 ± 8.2	2.6	0.002
High Density Lipoprotein (mg/dL)	42.4 ± 11.3	47 ± 11.9	10.8	<0.001
Triglycerides (mg/dL)	130 (100, 180)	110 (105, 140)	15.4	0.001
Hemoglobin A1C (%)	6.8 (5.7, 7.3)	6.5 (5.7, 7)	4.4	0.81

* Variables are summarized as the mean ± standard deviation or median (IQR).

**Table 3 jcm-10-04308-t003:** Secondary outcomes analysis.

Secondary Outcomes	At 6 Months Post-Implant
% Improvement in eGFR †	18.4%
Change in CKD Class: †† (a) Worsening (b) Improvement	7% (2/29) 41% (12/29)
% Death/Dialysis	0%
% Re-Admission rate	17% (5/29)
% Improvement in NYHA Class †††	31% (9/29)
% Improvement in BNP ††††	55.7%

† eGFR improved significantly from baseline (*p* = 0.04). †† CKD class improved significantly from baseline (*p* = 0.008). ††† NYHA class improved significantly from baseline (*p* = 0.003). †††† BNP improved significantly from baseline (*p* = 0.002).

## Data Availability

The datasets analyzed in study are available from the corresponding author on reasonable request.

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
