# Peer review of "Improvement in Metabolic Co-Morbidities after Implantation of CardioMEMS in Patients with Heart Failure with Preserved Ejection Fraction Phenotype"

_jcm, 2021, doi:10.3390/jcm10194308_

Round 1
Reviewer 1 Report
The present paper explored the effect of hemodynamic monitoring via CardioMEMS on comorbidities associated with the metabolic syndrome in patients with heart failure with preserved ejection fraction (HFpEF). In this retrospective study, a small sample of 29 patients were consecutively analyzed. Six months after device implantation, pulmonary artery diastolic pressure (PADP) was improved alongside several other metabolic parameters, despite no change in medication beyond titration of diuretics. Overall, the paper is well-written and the results contribute to HFpEF patient management. There are a few considerations that may help improve the paper.
General Comments
- How many patients had pulmonary hypertension (PH)? The elevated PADP seems to suggest the presence of PH in at least some patients. Further, was there a correlation between baseline PADP and the change in the PADP at 6 months?
- What was the duration of time since HFpEF diagnosis? What was the duration of time since metabolic syndrome diagnosis?
- The present patient population had a BMI classification of severe obesity. How were potential respiratory variations in the pulmonary artery waveform addressed when obtaining CardioMEMS PADP measures?
- With no change in medications for hyperlipidemia, hypertriglyceridemia or diabetes from baseline to 6 months, can the authors speculate why HDL and TGL improved and HbA1c did not?
- How do the authors see the results from their work aligning with weight management and physical activity when treating metabolic syndrome?
Specific Comments
- Page 1, lines 26-27: As written, it seems like the CardioMEMS device controls PADP. Rather, this device simply provides a reading of pulmonary artery pressures. Please rephrase for additional clarity.
- Page 2, line 53: Where (which institution) was the data collected? Perhaps add the information from page 5, lines 168-169 to the methods for clarity.
- Page 2, lines 61-64: Information related to primary and secondary endpoints should be moved from the methods to the end of the introduction. This will help clearly form the direction of the paper.
- Throughout paper: ‘HgA1C’ should be changed to ‘HbA1C’ to align with common naming convention.
- Page 5, line 160: ‘HA1C’ should be updated accordingly.
- Page 5, line 166-167: Correlational analysis was not completed and this should be modified to reflect the analyses performed.
- Pages 4-5, lines 124-148: This aspect of the discussion reads a little like a review of literature and can be modified to better incorporate the findings from the current study.
- Table 1: Please provide appropriate variable units were applicable (e.g. creatine, eGFR, BNP).
Author Response
Dear editor and reviewer,
Thank you for giving us the opportunity to re-submit a revised draft of our manuscript titled “Improvement in Metabolic Co-Morbidities After Implantation Of CardioMEMS In patients with Heart Failure with Preserved Ejection Fraction Phenotype” to Journal of Clinical Medicine. We appreciate the time and effort that you and the reviewers have dedicated to providing your valuable feedback on our manuscript. We are grateful to the reviewers for their insightful comments on our paper. We have been able to incorporate changes to reflect most of the suggestions provided by the reviewers. We have highlighted the changes within the manuscript.
Here is a point-by-point response to the reviewers’ comments and concerns.
Reviewer #1:
General Comments
- How many patients had pulmonary hypertension (PH)? The elevated PADP seems to suggest the presence of PH in at least some patients. Further, was there a correlation between baseline PADP and the change in the PADP at 6 months?
- We appreciate the concern raised by the reviewer. 83% of patients (24/29) had a diagnosis of pulmonary hypertension (with mean PA of 31 mmHg) based on the index right heart catheterization. We have included this baseline characteristic in table 1 of our manuscript. Despite improvement of mean pulmonary pressure improvement at 6 months, we chose not to include this variable in Table 2 because our treatment algorithm was based off of PADP values.
- What was the duration of time since HFpEF diagnosis? What was the duration of time since metabolic syndrome diagnosis?
- We realize the importance of the concern raised by the reviewer. However, we are a tertiary referral center for many of the community and hospital cardiologists and unable to generalize the time frame of the onset of HFpEF or of the metabolic syndrome. Our study highlights that regardless of timing of diagnosis, with improvement in hemodynamic parameters, the comorbidities associated with the metabolic syndrome improved.
- The present patient population had a BMI classification of severe obesity. How were potential respiratory variations in the pulmonary artery waveform addressed when obtaining CardioMEMS PADP measures?
- We appreciate the point that is raised by the reviewer. Our wave form tracings and readings at end-expiration were collected. We have added this in our methods section.
- With no change in medications for hyperlipidemia, hypertriglyceridemia or diabetes from baseline to 6 months, can the authors speculate why HDL and TGL improved and HbA1c did not?
- We agree with the reviewer that this finding deserves a further explanation. We have included possible explanations in our limitation and future direction section.
- How do the authors see the results from their work aligning with weight management and physical activity when treating metabolic syndrome?
- We thank the reviewer for this question. Current ACC/AHA/ESC guidelines all endorse lifestyle modifications and support healthy habits such as physical activity to help improve outcomes in patients with heart failure. All of our patients with and without cardiomems were given recommendations as per the guidelines. We would expect a synergistic result in improvement of the metabolic syndrome as we improve the PADP however in the absence of a control group, our results are hypothesis generating. We have added this important limitation and quote, “The findings in our study are hypothesis generating in the absence of a control group.”
Specific Comments
- Page 1, lines 26-27: As written, it seems like the CardioMEMS device controls PADP. Rather, this device simply provides a reading of pulmonary artery pressures. Please rephrase for additional clarity.
- We appreciate the reviewer’s comment. We have removed the CardioMEMS from this statement and rephrased to state, “The utilization of CardioMEMS to optimize PADP results in improvement in the comorbidities associated with the metabolic syndrome.”
- Page 2, line 53: Where (which institution) was the data collected? Perhaps add the information from page 5, lines 168-169 to the methods for clarity.
- We have added the institution in our methods section. “This study was approved by the St. Francis Hospital’s institutional review board, and a requirement for informed consent was waived due to the retrospective nature of the study.
- Page 2, lines 61-64: Information related to primary and secondary endpoints should be moved from the methods to the end of the introduction. This will help clearly form the direction of the paper. Johanna
- Thank you for this suggestion. We agree that this change will help improve the flow of the manuscript and have made the changes accordingly.
- Throughout paper: ‘HgA1C’ should be changed to ‘HbA1C’ to align with common naming convention. Page 5, line 160: ‘HA1C’ should be updated accordingly.
- We thank the reviewer for this important error. We have updated HgA1C and HA1C to just HbA1C.
- Page 5, line 166-167: Correlational analysis was not completed and this should be modified to reflect the analyses performed.
- Thank you for pointing this out, we updated the language to “improvements from baseline to six months post-CardioMEMS implantation” instead of “correlations.”
- Pages 4-5, lines 124-148: This aspect of the discussion reads a little like a review of literature and can be modified to better incorporate the findings from the current study.
- We thank the reviewer for this insightful suggestion. We have modified the discussion and limitations/future directions to focus more on our study.
- Table 1: Please provide appropriate variable units were applicable (e.g. creatine, eGFR, BNP). Johanna
- We thank the reviewers for highlighting this missing detail. We checked the manuscript and added units for all the variables in Table 1.
Reviewer 2 Report
The authors suggest there is an improvement in metabolic co-morbidities after implantation of CardioMEMS in patients with HFpEF based on a retrospective analysis of 29 patients with HFpEF. However, although variables were assessed at baseline and six months post implant there is no control group, so these changes cannot be attributed to CardioMEMS, also the pathophysiology that could explain an improvement in the comorbidities associated with the metabolic syndrome due to CardioMEMS is not clear.
Author Response
Dear editor and reviewer,
Thank you for giving us the opportunity to re-submit a revised draft of our manuscript titled “Improvement in Metabolic Co-Morbidities After Implantation Of CardioMEMS In patients with Heart Failure with Preserved Ejection Fraction Phenotype” to Journal of Clinical Medicine. We appreciate the time and effort that you and the reviewers have dedicated to providing your valuable feedback on our manuscript. We are grateful to the reviewers for their insightful comments on our paper. We have been able to incorporate changes to reflect most of the suggestions provided by the reviewers. We have highlighted the changes within the manuscript.
Here is a point-by-point response to the reviewers’ comments and concerns.
Reviewer 2:
The authors suggest there is an improvement in metabolic co-morbidities after implantation of CardioMEMS in patients with HFpEF based on a retrospective analysis of 29 patients with HFpEF. However, although variables were assessed at baseline and six months post implant there is no control group, so these changes cannot be attributed to CardioMEMS, also the pathophysiology that could explain an improvement in the comorbidities associated with the metabolic syndrome due to CardioMEMS is not clear.
- We are very grateful for the time spent on reviewing our manuscript. We appreciate the concerns raised by the reviewer, namely a lack of a control group and lack of a proposed mechanism to explain the mechanism of action behind observed with the improvement in pulmonary artery pressures. We have updated our discussion and limitations sections in our manuscript to address both concerns.
Reviewer 3 Report
The study is interesting but some important data needed for correct interpretation is missing.
From previous studies on CardioMEMS we know that the dose of diuretics are most frequently modyfied. Could you provide information on drug changes following implantation and how these changes were correlated with metabolic profiele trajectories.
The rate of rehospitalisation is presented and claimed to be low. I would like to have information on rehospitalisations in a group of similar patients but treated without CardioMems
Author Response
Dear editor and reviewer,
Thank you for giving us the opportunity to re-submit a revised draft of our manuscript titled “Improvement in Metabolic Co-Morbidities After Implantation Of CardioMEMS In patients with Heart Failure with Preserved Ejection Fraction Phenotype” to Journal of Clinical Medicine. We appreciate the time and effort that you and the reviewers have dedicated to providing your valuable feedback on our manuscript. We are grateful to the reviewers for their insightful comments on our paper. We have been able to incorporate changes to reflect most of the suggestions provided by the reviewers. We have highlighted the changes within the manuscript.
Here is a point-by-point response to the reviewers’ comments and concerns.
Reviewer 3:
The study is interesting but some important data needed for correct interpretation is missing.
From previous studies on CardioMEMS we know that the dose of diuretics are most frequently modified. Could you provide information on drug changes following implantation and how these changes were correlated with metabolic profiele trajectories.
- Thank you for this interesting suggestion. Unfortunately, metabolic parameters were checked routinely prior to CardioMEMS implantation and at the 6 month mark. We do not have metabolic trajectories in between the 6-month time frame to assess for trajectories for additional time points between the ones we report in our manuscript. Furthermore, we have explained our frequency of diuretics medication titration in our medication usage section. “Diuretic adjustments, mostly in the form of loop diuretics were made 1-2 times per week on average for the first 6-8 weeks following implantation. After this initial period, loop diuretics were adjusted 0-1 times per week for the remainder of the study duration.”
The rate of rehospitalisation is presented and claimed to be low. I would like to have information on rehospitalisations in a group of similar patients but treated without CardioMems"
- Thank you for this interesting suggestion. Unfortunately, this is a retrospective analysis and we do not have a control group to compare our outcomes to. Historically, the CHAMPION trial has reported “In 6 months, 84 (32%) heart-failure-related hospitalizations were reported in the CardioMEMs group (n=270) compared with 120 (44%) in the control group (n=280; rate 0·32 vs0·44, hazard ratio [HR] 0·72, 95% CI 0·60–0·85, p=0·0002).” Our lack of control group and that our findings are hypothesis generating is now included in the limits and future directions section. We quote “The findings in our study are hypothesis generating in the absence of a control group.”
Round 2
Reviewer 2 Report
My previous comments remain.